# Prevalence of Coronary Artery Calcification on Pre-Atrial Fibrillation Ablation CT Pulmonary Venograms and its Impact on Selection for Statin Therapy

**DOI:** 10.3390/jcm9061631

**Published:** 2020-05-28

**Authors:** Michael P. Dunleavy, Avirup Guha, Andrea Cardona, Christopher Fortuna, Emile G. Daoud, Subha V. Raman, Thura T. Harfi

**Affiliations:** 1Division of Hospital Medicine, The Ohio State University Wexner Medical Center, Columbus, OH 43210, USA; michael.dunleavy@osumc.edu; 2Division of Cardiovascular Medicine, The Ohio State University Wexner Medical Center, Columbus, OH 43210, USA; guha23@osu.edu (A.G.); andrea.cardona@osumc.edu (A.C.); emile.daoud@osumc.edu (E.G.D.); suraman@iu.edu (S.V.R.); 3Division of Cardiovascular Medicine, Harrington Heart and Vascular Institute, Cleveland, OH, 44106, USA; 4The Ohio State University College of Medicine, Columbus, OH 43210, USA; christopher.fortuna@osumc.edu

**Keywords:** CT pulmonary venogram, coronary artery calcium, atherosclerosis, atrial fibrillation, statin therapy

## Abstract

Atherosclerotic cardiovascular disease (ASCVD) shares many risk factors with atrial fibrillation (AF). Obtaining computed tomography images of the pulmonary veins (CTPV) before AF ablation procedures is common and can incidentally detect coronary artery calcification (CAC). The purpose of this study was to investigate the prevalence of CAC on pre-ablation CTPV, the frequency of CAC reporting on CTPV reports, and its impact on statin therapy among patients hospitalized for AF procedures. We retrospectively evaluated consecutive patients undergoing CTPV and AF procedures from October 2016 to December 2017 in a single-center tertiary hospital. The patients’ demographic and clinical characteristics were analyzed. The CAC presence on CTPV was visually assessed. The severity was classified qualitatively. The statin therapy status was evaluated using the patient’s admission and discharge medication lists. A total of 638 subjects were included in our study, with 34.5% female. The mean age was 63.3 ± 10.8 years. CAC was detected in 70.1% of all patients, and in 58.1% of patients without a history of ASCVD. When present, CAC was documented in 92.6% of the clinical CTPV reports. While coronary artery atherosclerosis was present in a majority of AF patients, and its presence was widely reported, it was not associated with increased statin therapy at discharge.

## 1. Introduction

Atrial fibrillation (AF) is the most common arrhythmia affecting the aging American population, with an estimated prevalence of 12.5 million by 2030 [1]. Treatment of AF using pulmonary vein (PV) isolation, also known as AF ablation, has been increasing sharply over the past decade [2,3,4]. Most patients undergoing AF ablation undergo contrast-enhanced computed tomography (CT) of the pulmonary veins (CTPV) for procedural planning. Since CTPV includes the coronary arteries in the field of view, the presence of coronary artery calcification (CAC) can be easily detected. CAC is an important marker of clinical and subclinical atherosclerotic cardiovascular disease (ASCVD) and a strong independent predictor of major adverse cardiovascular outcomes [5,6,7,8,9,10,11]. While CAC is typically detected and quantified using a dedicated non-contrast CT CAC scan with electrocardiogram(ECG)-gating, CAC detected using contrast-enhanced cardiac CT correlates strongly with dedicated CAC scans [12,13,14,15]. Evaluating and reporting CAC on all non-contrast chest CT scans is recommended, though it is not commonly reported [16,17,18,19]. CAC was incorporated into the 2013 guidelines as one of the few other tests that function as a risk modifier in subjects at intermediate risk and was explicitly incorporated into the CV risk prediction guidelines in 2018 [20,21].

Currently, the prevalence of CAC on CTPV and its reporting is unknown. Moreover, the missed opportunity to therapeutically prevent ASCVD-related events using statin therapy in AF patients, who share many of the same risk factors, has never been quantified. The purpose of this study was to investigate the prevalence of CAC on pre-ablation CTPV, the frequency of CAC reporting on clinical CTPV reports, and the potential impact of CAC reporting on patient selection for statin therapy among patients hospitalized for AF-related procedures.

## 2. Materials and Methods

We retrospectively selected consecutive patients who underwent clinically indicated CTPV and AF ablation or left atrial appendage occluder device (LAAOD) placement procedures from October 2016 to December 2017. Patient demographics and clinical variables were analyzed. Indications for statin therapy for primary and secondary prevention of atherosclerotic disease were considered in accordance with the 2013 American College of Cardiology/American Heart Association (ACC/AHA) guidelines on the treatment of blood cholesterol, which were the most current guidelines at the time of admission [20]. Patients with a history of clinical ASCVD were classified as having a history of coronary artery disease (CAD), peripheral artery disease, carotid artery disease, transient ischemic attack, or stroke, and were considered candidates for secondary prevention. We considered all strokes and transient ischemic attacks (TIAs) to be atherosclerotic since we were unable to determine which strokes and TIAs were atherosclerotic versus embolic in origin. Patients without a history of clinical ASCVD were considered candidates for primary prevention, where their 10-year ASCVD risk was calculated using the ACC/AHA ASCVD Pooled Cohort Risk Calculator [22]. We additionally evaluated the status of statin therapy for all patients admitted to the hospital using their admission medication list.

CTPV image acquisition was performed using a Discovery CT750 HD 64-slice CT scanner (GE Healthcare, Chicago, IL, USA) (field of view (FOV) 200 × 200 mm, slice thickness 0.63 cm, 120 kV as determined by the CT scanner). CTPV scanning was contrast-enhanced and performed with or without ECG-gating depending on the patient rhythm and date of the study. Before 15 May 2017, all patients underwent non-ECG-gated spiral acquisition. After that date, prospective ECG-gating was introduced and used for patients who had a normal sinus rhythm, while patients who were in AF at the time of the scan continued to have non-ECG-gated spiral acquisition. Such a change was due to a clinical change in the image acquisition practice of the cardiac CT lab unrelated to our research study. CAC presence on CTPV images was visually assessed by an expert reader (TTH) with 5 years of experience in cardiac CT. The reader was completely blinded to the clinical data. The intraobserver correlation was not assessed. In a similar pattern to prior literature, the CAC severity was classified visually in a qualitative manner into mild, moderate, or severe [16,23] (Figure 1). CAC was considered mild if calcification involved a short segment or a focal area of a single coronary artery, and was considered severe if calcification was dense and involved long segments of multiple coronary arteries. CAC was classified as moderate if there was more CAC than could be considered mild but it was less than the description of severe CAC. Patients with coronary stents were considered to have severe CAC. We evaluated the frequency of reporting CAC status on clinical CTPV reports. The prevalence and severity of CAC were calculated in the total study population, as well as in different statin therapy groups. Additionally, we analyzed the impact of reporting the CAC status in CTPV reports on statin therapy modification by comparing the discharge medication list with the medication list at admission. This study was approved by The Ohio State University institutional review board.

Data are presented as mean ± standard deviation (SD) for continuous variables and as proportions for categorical variables. A paired *t*-test was used to compare the mean values of continuous variables. An ANOVA test was used to compare mean values of more than two continuous variables. A Fisher test was used to compare statistical differences between the proportions of different categories. Statistical significance was set at two-tailed *p* < 0.05. IBM SPSS Statistic 21.0 (Chicago, IL, USA) was used for all statistical analyses.

## 3. Results

### 3.1. Baseline Population

A total of 638 patients were included in the study, where the mean age was 63.3 ± 10.8 years old. A total of 34.5% of the subjects were female and 95% were white. Forty percent were former smokers, and 6.9% were current smokers. Over three-quarters (77.7%) had hypertension and 23.4% had diabetes. The average BMI was 32.9. Most patients (71.8%) underwent non-ECG-gated spiral CT acquisition. The vast majority of patients (95.6%) underwent AF ablation; 1.6% underwent LAAOD placement; and the rest had their AF ablation procedure canceled, rescheduled, or replaced with a cardioversion procedure. Prior ASCVD was present in 252 (39.6%) of the patients. Stroke and/or TIA were seen in 66 patients (26.2%) with prior ASCVD, of which CAD was co-existent in 36 patients (55%). Among patients with prior ASCVD, only 30 (11.9%) patients had a stroke and/or TIA without CAD. The demographic and clinical characteristics are summarized in Table 1. The available data allowed for the calculation of the 10-year ASCVD risk for 290 subjects (45.5% of the total population). Among patients without prior ASCVD, 79.3% had an elevated 10-year ASCVD risk of ≥5% (Table 1).

### 3.2. CAC Findings on CTPV

CAC was detected on CTPV in 70.1% of patients. CAC was more prevalent in patients with prior ASCVD (88.1%) compared to those without (58.1%) (*p* < 0.0001). Among the 66 patients with stroke and/or TIA, CAC was also very prevalent, as it was detected in 51 patients (77.3%). In patients without a history of clinical ASCVD, mild, moderate, and severe CAC was detected in 34.1%, 16.4%, and 7.6%, respectively (Table 1). The 10-year ASCVD risk correlated with the presence of CAC. The average ASCVD risk among those with no CAC, mild CAC, moderate CAC, and severe CAC was 10.7%, 14.8%, 18.2%, and 22.1%, respectively (*p* < 0.001, Figure 2). CAC was highly prevalent in the primary prevention group. Among the borderline and intermediate-risk (5%–20% ASCVD risk) patients, 61% had CAC, while 76% of those with high risk (>20% ASCVD risk) had CAC. (Figure 3). CAC presence was documented on the final clinical CTPV reports for 92.6% of patients who had CAC.

### 3.3. Statin Therapy Details

Statin therapy was more prevalent in patients with prior ASCVD (80.1%) than in those without (34.7%) (*p* < 0.0001). Among those with prior ASCVD, 19.9% were not on any statin therapy and only 38.6% were on a high-intensity statin (Table 1). Statins were also under-prescribed in the primary prevention population. Only 21.6% of those considered high risk (>20% 10-year risk) and 43.1% of those considered intermediate risk (7.5%–20% 10-year risk) were on statins at admission (Figure 4). Overall, statin therapy was only prescribed in 38% of the statin-eligible patients without a history of ASCVD (i.e., those with a 10-year ASCVD ≥5%). A similar pattern appeared when evaluating statin therapy based on CAC detection in the primary prevention population. Only 41.1% of those who had CAC on CTPV were on statin therapy. Additionally, about half (48.4%) of those who had moderate or severe CAC were not on statin therapy. Meanwhile, 26.1% of those who did not have CAC were on statin therapy (Figure 4). The majority (58.6%) of the primary prevention group had a borderline or intermediate ASCVD risk of 5%–20%, and 61% of this group had CAC. Over half of those who were borderline or intermediate risk and had CAC (51.6%) were not on statin therapy. CAC resulted in significant improvements in risk assessment among the primary prevention group with borderline and intermediate risks. The calculated net reclassification index (NRI) was 46%, reflecting those with borderline or intermediate risk who were on a statin but did not have CAC detected or were not on a statin but had CAC present. The calculated NRI for the primary prevention group with a low risk (< 5%) was 35% (Figure 3).

### 3.4. Statin Prescription on Discharge

There was a minimal change in the statin prescription at discharge for those with a history of ASCVD. Of the 251 patients with ASCVD, 154 (61.3%) were undertreated with statin (no statin or not on a high-intensity statin). There were 70 patients (64.2%) with no history of ASCVD but an elevated 10-year ASCVD risk of ≥7.5% who were not on statin therapy, none of whom were started on a statin at the time of discharge.

Additionally, there were 50 patients with a history of ASCVD who were not on statin therapy at admission, and only 2 (4%) of those patients were started on a statin. One patient with ASCVD had the statin intensity increased from low to medium. Additionally, in those without ASCVD, the detection of CAC on CTPV did not result in any statin prescription at discharge. Among those without ASCVD who were not on statin therapy at admission, CAC was detected in 131 patients, with 44 of them having moderate or severe CAC (Figure 4). None of these patients were prescribed a statin upon discharge.

## 4. Discussion

Our study showed that in patients who were admitted to a major tertiary center primarily for an AF ablation, CAC was highly prevalent and detected in 70.1% of the CTPV scans and reported in the vast majority (92.8%) of CTPV clinical reports. Our study also shows that there was significant underutilization of statins in AF patients and that reporting CAC in CTPV reports did not improve the statin therapy pattern. The reporting rate of CAC on CTPV in our study of 92.8% was higher than in prior studies showing the reporting of CAC to be 18.6%–69% on CT clinical reports [17,18,19]. This excellent reporting ratio can be explained by the fact that all CTPVs in this study were read by cardiologists who were trained in cardiac CT and aware of the clinical importance of CAC detection. Prior studies have shown that cardiac imagers are more likely to report CAC than non-cardiac imagers [24].

Although clinical ASCVD was present in only 39.6% of the study population, coronary atherosclerosis (detected as CAC) was much more common at 70.1%. The high prevalence of CAD (55%) and CAC (77%) in patients who had a stroke and/or TIA is likely related to shared risk factors between AF and CAD and a potential selection bias as our study population represented only the subgroup of AF patients who were eligible for an AF ablation. It is noteworthy that CAC was present in the majority (58.1%) of patients without known ASCVD. Our population’s incidence of documented clinical ASCVD of 39.6% was comparable to that found in prior studies [25,26]. There is a high cardiovascular event rate in patients with AF [27,28]. The high prevalence of CAC in our study population might help to explain the high cardiovascular event rate in AF patients.

Further, our study showed statins were significantly underutilized in the study population for both primary and secondary prevention purposes. About one-fifth of patients with a history of ASCVD were not on a statin at admission and a high-intensity statin was only seen in about one third (38.6%) of patients. Among patients without known ASCVD, statin therapy was noted in 21.6% of high-risk patients and 43.1% of intermediate-risk patients. The problem of statin under-utilization is widespread, including in those with secondary prevention indications for statin therapy. While high-intensity statins are recommended for all patients with CAD, multiple studies in patients with prior known CAD have shown only 55%–77% were on any statin therapy [29,30,31]. It has been shown that the detection of CAC is an effective motivator for patients to improve their lifestyle and improve compliance with CV preventive therapies, such as aspirin and statins [32,33]. Patients who are at high ASCVD risk but hesitant to start statin therapy might be motivated once they are informed about having CAC [34,35]. Furthermore, our study shed light on a significant missed opportunity for statin optimization, especially in the primary prevention group, where 58.9% had CAC but were not on a statin. This high CAC prevalence in the primary prevention group was also seen in those with an ASCVD risk of 5%–20%, where CAC status was highly impactful on the statin therapy decision. Moreover, almost half of the primary prevention patients with moderate or severe CAC were not on statin therapy. Despite admission to a cardiology service after undergoing an AF ablation procedure, there was minimal statin therapy adjustment upon discharge. Moreover, reporting the presence of CAC on CTPV reports did not seem to promote statin therapy adjustments.

There are several limitations to our study. It was not designed to investigate the reasons for the lack of statin prescription adjustment in response to CAC detection on CTPV scans, though they were likely multifactorial. Patients undergoing AF ablation were admitted for short hospital stays, where the focus was the detection of procedural complications and early discharge rather than the optimization of preventative medication therapy. Additionally, most CTPV reports did not include a quantitative assessment of CAC; therefore, no specific statin-related therapeutic recommendations were included in the report. Furthermore, while it is widely accepted that CAC presence is associated with a higher ASCVD risk, and was incorporated into the 2013 guidelines as one of the several other tests that functions as a risk modifier in subjects at intermediate risk, it was not explicitly incorporated into the CV risk prediction guidelines until after the time of our study in 2018 [21,22]. Furthermore, our study was retrospective and the data was collected via chart reviews, where complete information about the variables of interest was not available for all the study population. Due to this, we were unable to distinguish the origin of stroke or TIA as atherosclerotic or cardioembolic. The study included primarily white patients undergoing an AF ablation or left atrial appendage closure procedure, which limits its applicability to the overall AF population and to non-white patients. The generalizability of our results is limited considering the study is single-centered, where CTPV were exclusively read by cardiologists. Finally, CAC was qualitatively assessed via visual assessment from a contrast-enhanced CT image. However, this was also a strength since by using CTPV images to extract CAC information is preferred to exposing the patient to additional radiation for a dedicated CAC scan.

## 5. Conclusions

Coronary artery atherosclerosis was detected in a majority of AF patients during pre-ablation CTPV and its presence was widely reported in clinical CTPV reports. The high reporting of CAC was not associated with increased statin therapy at discharge. CAC detection on pre ablation CTPV represents a large, missed opportunity to potentially optimize statin therapy and improve cardiovascular outcomes in AF patients.

## Figures and Tables

**Figure 1 jcm-09-01631-f001:**
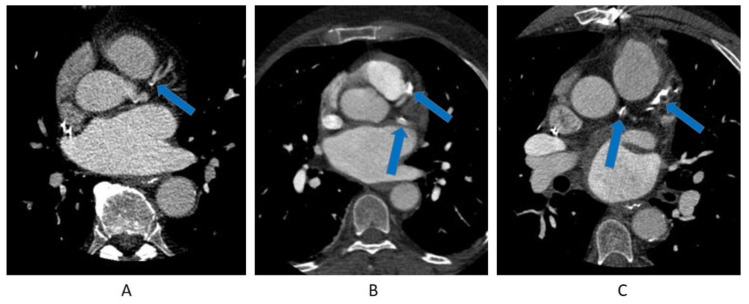
Coronary artery calcification detection on computed tomography (CT) pulmonary venograms. (**A**) Mild coronary calcification, seen as a spot of calcification in the proximal left anterior descending artery. (**B**) Moderate coronary calcification, seen as a bulky calcification in the proximal part of the left anterior descending artery. Another area of calcification is seen in the proximal part of the left circumflex artery. (**C**) Severe coronary calcification, seen as a very dense and long segment of the proximal and mid-portion of the left anterior descending artery. Additional coronary calcification is seen involving the left main coronary artery.

**Figure 2 jcm-09-01631-f002:**
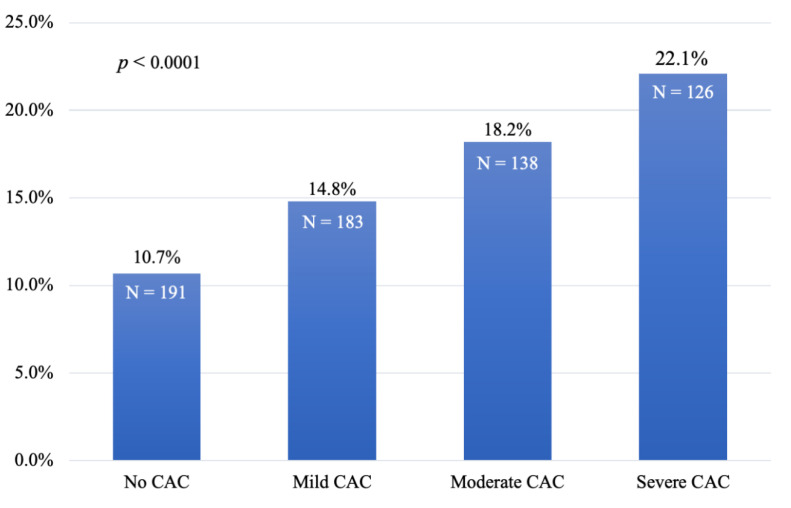
Association of the CAC grade with the 10-year risk of ASCVD. The average 10-year ASCVD risk score of patients in the total study population was based on their respective CAC classification of no CAC, mild CAC, moderate CAC, or severe CAC. *p* < 0.0001. CAC—coronary artery calcification, ASCVD—atherosclerotic cardiovascular disease.

**Figure 3 jcm-09-01631-f003:**
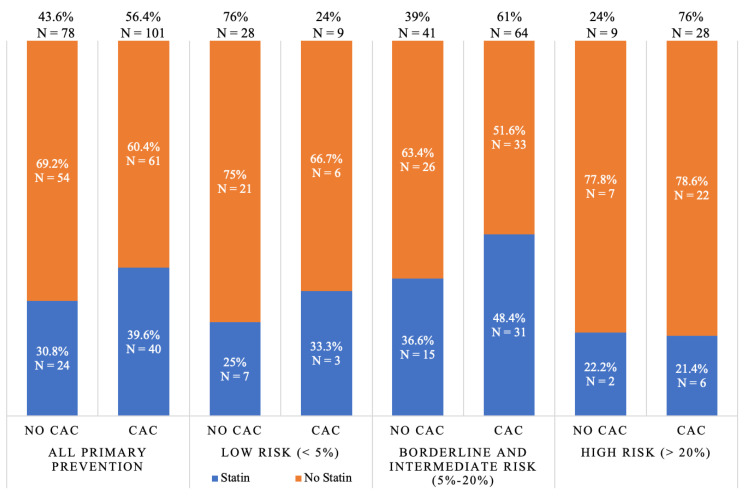
Prevalence of statin therapy at the time of admission relative to the presence or absence of CAC and ASCVD risk in the primary prevention group. Data limited to the 179 patients for which ASCVD risk and statin therapy information were available. CAC—coronary artery calcification, ASCVD risk—atherosclerotic cardiovascular risk score by the Pooled Cohort Risk Calculator.

**Figure 4 jcm-09-01631-f004:**
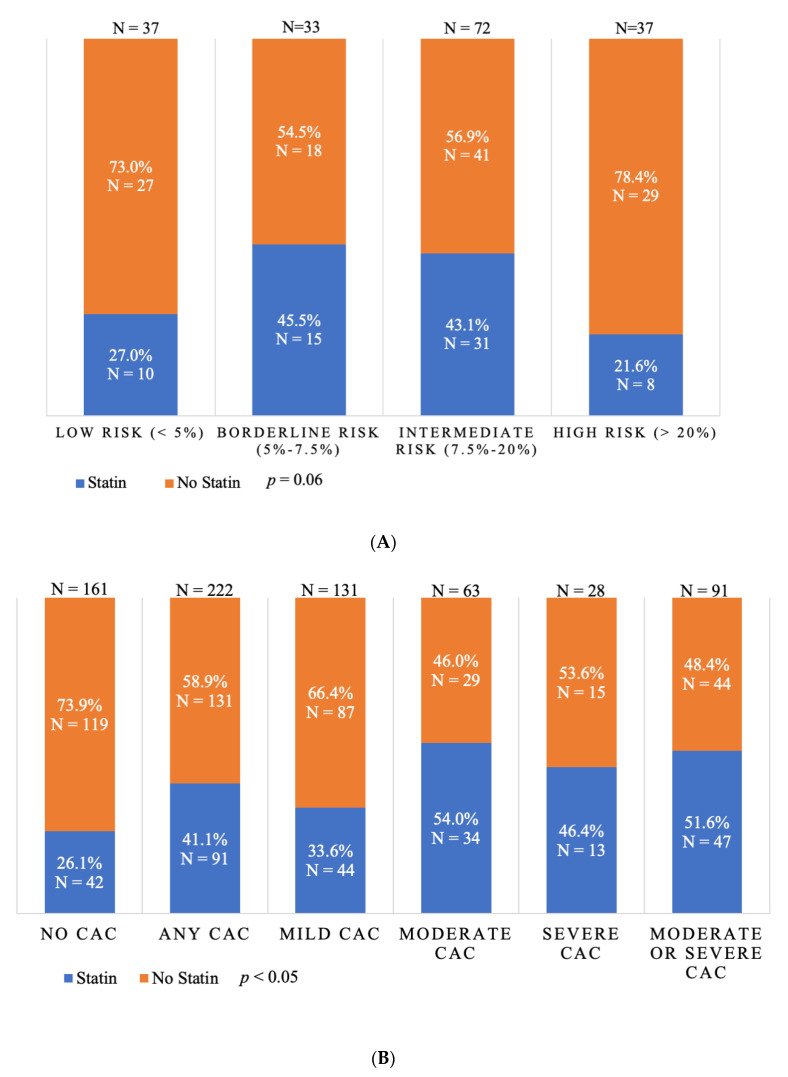
(**A**) Statin therapy use based on 10-year ASCVD risk and statin therapy distribution across different levels of 10-year ASCVD risk among subjects without ASCVD, *p* = 0.06. (**B**) Statin therapy distribution across different levels of CAC on CTPV among patients without a history of ASCVD, *p* < 0.05. CAC—coronary artery calcification, CTPV—computed tomography of the pulmonary veins, ASCVD—atherosclerotic cardiovascular disease

**Table 1 jcm-09-01631-t001:** Demographic and clinical characteristics of the study population*.

Characteristics	Total Population	No Prior ASCVD	Prior ASCVD	*p*-Value
Population, N	638	384 (60.4)	252 (39.6)	
Age, years	63.3 ± 10.8	61.3 ± 11.7	66.3 ± 8.6	<0.0001
Female, N (%)	220 (34.5)	140 (63.6)	80 (36.4)	0.22
Race (white), N (%)	606 (95)	368 (60.8)	237 (39.2)	0.48
Body mass index kg/m^2^	32.9±12.9	33.4 ±15.7	32.4 ± 7.3	0.72
CT scanning mode, N (%)				
Non-ECG-gated, spiral	455 (71.8)	261 (68.1)	194 (77.3)	0.01
Prospective ECG-gated	179 (28.2)	122(31.9)	57 (22.7)	
Smoking status, N (%)				<0.0001
Never	335 (52.8)	228 (59.7)	105 (42)	
Current	44 (6.9)	28 (7.3)	16 (6.4)	
Former	255 (40)	126 (33)	129 (51.6)	
Cardiovascular History, N (%)				
Hypertension	496 (77.7)	266 (53.9)	228 (46.1)	<0.0001
Diabetes mellitus	149 (23.4)	65 (43.9)	83 (56.1)	<0.0001
Baseline Cardiovascular Examination				
LVEF, %	54.7 ± 9	55.9 ± 7.6	53.0 ± 10.4	0.0002
EF < 50%, N (%)	117 (18.8)	52 (13.9)	64 (25.7)	0.0002
Baseline Labs				
Creatinine, mg/dL	1.01 ± 0.46	0.95 ± 0.2	1.09 ± 0.67	0.03
LDL-C, mg/dL	88.9 ± 35	97.4 ± 39.9	78.8 ± 36.9	<0.0001
ASCVD History, N (%)				
Clinical CAD	219 (34.4)		219 (86.9)	
PAD	12 (1.9)		12 (4.8)	
Stroke	38 (6)		38 (15.1)	
TIAStroke and/or TIA	47 (7.4)66 (10.3)		47 (18.7)66 (26.2)	
Carotid Artery Disease	21 (3.3)		21 (8.3)	
10-year ASCVD Risk Score, N (%) **				
Total	15.3 ± 12.1	12.9 ± 10.4	19.3 ± 13.7	<0.0001
Low risk < 5%	45 (15.5)	37 (20.7)	8 (7.3)	
Borderline risk 5% to < 7.5%	41 (14.1)	33 (18.4)	8 (7.3)	
Intermediate risk 7.5% to 20%	131 (45.2)	72 (40.2)	58 (52.7)	
High risk ≥ 20%	73 (25.2)	37 (20.7)	36 (32.7)	
Statin therapy on admission, N (%)				
YES	336 (52.8)	133 (34.7)	201 (80.1)	<0.0001
NO	300 (47.2)	250 (65.3)	50 (19.9)	
Statin intensity, N (%)				
High intensity	131 (20.6)	32 (8.4)	97 (38.6)	<0.0001
Moderate intensity	179 (28.2)	87 (22.8)	92 (36.7)	
Low intensity	25 (4.0)	13 (3.4)	12 (4.8)	
CAC Detection, N (%)				
CAC Positive	447 (70.1)	223 (58.1)	222 (88.1)	<0.0001
CAC Negative	191 (29.9)	161 (41.9)	30 (11.9)	
CAC Grading, N (%)				
0 (none), N (%)	191 (29.5)	161 (41.9)	30 (11.9)	<0.0001
1 (mild), N (%)	183 (28.7)	131 (34.1)	51 (20.2)	
3 (moderate), N (%)	138 (21.6)	63 (16.4)	75 (29.8)	
5 (severe), N (%)	126 (19.7)	29 (7.6)	96 (38.1)	
CAC Reported, N (%) †	414 (92.6)	202 (90.6)	210 (94.6)	<0.0001

* Due to missing variables and rounding, values might not add up to complete percentages. ** Available for 290 subjects with 110 subjects had prior ASCVD. † CAC documented on CTPV report/total CAC detected. CAC—coronary artery calcification; CAD—coronary artery disease; PAD—peripheral artery disease; TIA—transient ischemic attack; LVEF—left ventricular ejection fraction; EF—ejection fraction; LDL-C—low-density lipoprotein cholesterol; ASCVD—atherosclerotic cardiovascular disease defined as a history of coronary artery disease (CAD), peripheral artery disease, carotid artery disease, transient ischemic attack, or stroke (patients with these diseases were considered candidates for secondary prevention). We considered all strokes and TIAs to be atherosclerotic in origin. ASCVD risk—atherosclerotic cardiovascular risk score calculated using the Pooled Cohort Risk Calculator.

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
