# Peer review of "Prevalence of Coronary Artery Calcification on Pre-Atrial Fibrillation Ablation CT Pulmonary Venograms and its Impact on Selection for Statin Therapy"

_jcm, 2020, doi:10.3390/jcm9061631_

Round 1

Reviewer 1 Report

General comments:

This retrospective study evaluated the prevalence of incidentally detected coronary artery calcification (CAC) on CTPV, the frequency of CAC reporting and its impact on statin therapy among patients hospitalized for AF procedures. CAC was present in 70% of all patients (n=638) and in 58% of those without history of atherosclerotic disease. CAC reporting was high (93%), but statin therapy at discharge was not increased.

The authors are commended for the elaborate work and the data provided. The manuscript is well written and interesting for the readers of the Journal. However, some issues need to be addressed before considering publication.

Specific comments:

  1. Methods: Did all patients undergo the same CCT scan protocol? Please report on the details of the protocol (parameters including kV and mA, beam collimation, slice thickness, field of view, breath hold etc.).
  2. Methods/Results: An expert reader reviewed the CCT scans (a lot of work!). Was he blinded to the clinical data of the patients? What is the inter- and intra-observer correlation for such measurements? Should there be any concern for observational bias? Please comment
  3. Methods/Results/Discussion: CAC score can reclassify risk up or down for patients with ASCVD of 5-7.5% and 7.5-20%. Since this study focuses in the value of CAC as incidental finding, consider reporting and commenting on the net reclassification improvement (NRI) of ASCVD alone vs. ASCVD+CAC. What percentage of patients would be reclassified to take or not take statins?

Reviewer 2 Report

The study from Dunleavy et Al shows that in patients who were admitted for an AF ablation, CAC was highly prevalent and detected in 70.1% of CTPV scans and reported in the vast majority (92.8%) of CTPV clinical reports. The study also shows that there is significant underutilization of statins in AF patients and that reporting CAC in CTPV reports did not improve the statin therapy pattern

This is an interesting result which underline the problem of underreporting the presence but also the absence of CAC in CT scan in general

Comments and Suggestions for Authors 

- P.3, line 87 - What is the meaning of “expert reader”? How many years of experience in cardiac CT?

- CAC severity classification is not clearly defined: CAC is considered mild if calcification involve a short segment or a focal area of a single coronary artery and is considered severe if calcification is dense and involve long segments of multiple coronary arteries but there is no definition for moderate CAC. Please clarify. CAC severity was classified visually but it differs from the two visually scores described by Chiles C et al [23]. Why the authors did not use one of the previously defined scores?

- It would be interesting to have the number of CTPV with and without prospective ECG gating.

- A qualitative CTPV score would have been interesting since coronary analysis can be difficult due to movement artifacts, particularly at the level of the right coronary artery, in non-ECG spiral gated acquisitions.

- What about patients with coronary stents? Have they been excluded or considered as severe CAC patients

-In a general population, over 20% of strokes occur in relation to atrial fibrillation (AF) and, for the great majority of these patients, secondary stroke prevention should be with anticoagulation. These strokes should not be included in atherosclerosis ischemic strokes and consequently in ASCVD. In a population of patients with AF, like in this study, the amount of strokes and of some TIA from cardio embolic origin is likely to be much higher. It would be interesting to make the difference in order

  1. to know precisely the percentage of pts with ASCVD in a cohort of AF referred for AF ablation. Certainly, less than 39.6%
  2. To know precisely the percentage of CAC in patients without clinical ASCVD referred for AF ablation. Certainly less than 30.5%

-In 2018 AHA/ACC/AACVPR/AAPA/ABC/ACPM/ADA/AGS/APhA/ASPC/ NLA/PCNA Guideline on the Management of Blood Cholesterol (JACC 2019) it is recommended to use CAC score for primary prevention in patient with intermediate risk (Class IIa). If the score is 0, statin should not be given unless cigarette smoking, diabetes or family history of premature CHD (p e307). This recommendation may be apply to the patients at low (25%) or intermediate risk (36.6%) showed in the interesting supplementary figure 1 and mentioned in the manuscript. The supplementary figure1 seems more informative than figure 3 and 4.

I suppose that absence of CAC is never reported in CTPV report for patients at low or intermediate risk and under statin. Could you confirm?

- Table 1: Correct or explain why in several lines of the table, the sum of the columns is not correct. In the first line, for example, the number of patients with ASCVD is 384 and without ASCVD is 252 while the total population is 638.

- Discussion: Is the population of the study representative of the general population? Indeed, 95% of the patients included were white and the average BMI was 32.9. As shown by Grundy SM et al [21], CAC score is highest in white and Hispanic men, with blacks having significantly lowerprevalence and severity of CAC. These two criteria can increase the prevalence of coronary calcifications in the population of the study.

Round 2

Reviewer 2 Report

The Net Reclassification Index introduced by the authors appears to be an interesting information about the usefulness of CAC (+or-) reporting for patients at borderline or intermediate risk.

But even if CAC score scanning in low risk patients is not recommended, once a CTPV is done, its findings (+ or -) when there are relevant according to the clinical situation should be reported. The NRI would be also interesting to mention in this population as some of these patients may have undue statin

In the guideline 2013 ACC/AHA Guideline on the Treatment of Blood Cholesterol to Reduce Atherosclerotic Cardiovascular Risk in Adults, we can read “For this guideline, ASCVD includes coronary heart disease (CHD), stroke, and peripheral arterial disease, all of presumed atherosclerotic origin” . I think that strokes occurring in patient with AF are not presumably atherosclerotic but result more probably from cardioembolism. Furthermore It is not proven by RCT that statin reduce total mortality in such case nor is useful in secondary stroke prevention.

Consequently I think it is not appropriate to classify patient with stroke and AF in the ASCVD group and the results of the study are not correct

Author Response

Dear Journal of Clinical Medicine Reviewers,

On behalf of all the authors, I would like to again thank the reviewer for their thought provoking questions, comments and recommendations. We appreciate their valuable time spent in reviewing our manuscript. We have addressed the reviewers’ comments as detailed below. For ease of reading, we have highlighted the reviewers’ comments with a bold font and kept our responses in a non-bold font.

The Net Reclassification Index introduced by the authors appears to be an interesting information about the usefulness of CAC (+or-) reporting for patients at borderline or intermediate risk.

But even if CAC score scanning in low risk patients is not recommended, once a CTPV is done, its findings (+ or -) when there are relevant according to the clinical situation should be reported. The NRI would be also interesting to mention in this population as some of these patients may have undue statin

We thank the reviewer for the comments. We have added the following sentence to the results section to provide further details about the NRI in low risk populations:

The calculated NRI for the primary prevention group with low risk (<5%) is 35%

In the guideline 2013 ACC/AHA Guideline on the Treatment of Blood Cholesterol to Reduce Atherosclerotic Cardiovascular Risk in Adults, we can read “For this guideline, ASCVD includes coronary heart disease (CHD), stroke, and peripheral arterial disease, all of presumed atherosclerotic origin” . I think that strokes occurring in patient with AF are not presumably atherosclerotic but result more probably from cardioembolism. Furthermore It is not proven by RCT that statin reduce total mortality in such case nor is useful in secondary stroke prevention.

Consequently I think it is not appropriate to classify patient with stroke and AF in the ASCVD group and the results of the study are not correct

We thank the reviewer for their well thought out comments. Unfortunately, due to the retrospective nature of our study, we were unable to tell what percentage of the patients with TIA and/or stroke were before or after they developed AF (thus being of atherosclerotic or cardioembolic in origin), and therefore we did not feel comfortable assuming they did not have an indication for statin therapy. Due to this challenge, we would be hesitant to assume the stroke was embolic and recommend discontinuation of statin therapy. Moreover, we felt including them in the primary prevention group may inappropriately inflate the primary prevention data since some of them may have had atherosclerotic stroke and/or TIA. Therefore, we felt it was most appropriate to keep them in the secondary prevention group. We have adjusted our methods section in the following was to make clear to our readers this distinction:

Patients with a history of clinical ASCVD were defined as a history of coronary artery disease (CAD), peripheral artery disease, carotid artery disease, transient ischemic attack, or stroke, were considered candidates for secondary prevention. We considered all strokes and TIAs to be atherosclerotic since we were unable to determine which strokes and TIAs were atherosclerotic versus embolic in origin.

Considering the challenge of where to place this specific population, we felt it would be appropriate to give additional information on these patients for the readers, so they could consider the impact on the results. These results showed that there was a total of 66 patients who had stroke and/or TIA. This accounted for 26.2% of the total secondary prevention group. Among these 66 patients, 36 (55%) patients had clinical CAD. That means that in the whole secondary prevention group, there were only 30 (11.9%) patients who had TIA or/and stroke and had no clinical CAD. Furthermore, the vast majority (77.3% or 51 patients) of the 66 patients with stroke&/or TIA had CAC on their CTPV

We added the following lines in the results:

Stroke and/or TIA were seen in 66 patients (26.2%) of those with prior ASCVD, of which CAD was co-existent in 36 patients (55%). Among patients with prior ASCVD, only 30 (11.9%) patients had stroke and/or TIA without CAD.

Among the 66 patients with stroke and/or TIA, CAC was also very prevalent as it was detected in 51 patients (77.3%).

We also updated the table to include the numbers of those with TIA and/or Stroke together as well as independent. We also made clear in the table that these patients are considered to be in the secondary prevention group.

ASCVD = atherosclerotic cardiovascular disease defined as a history of coronary artery disease (CAD), peripheral artery disease, carotid artery disease, transient ischemic attack, or stroke, were considered candidates for secondary prevention. We considered all strokes and TIAs to be atherosclerotic in origin.

We included our thoughts on this specific population in the discussion:

The high prevalence of CAD (55%) and  CAC (77%) in patients who had stroke and/or TIA is likely related to shared risk factors between AF and CAD and potential selection bias as our study population represents only the subgroup of AF patients who are eligible for an AF ablation.

Finally, since we do recognize it is a limitation that we cannot determine the cause of the stroke or TIA so we included the following in our limitations:

Also, our study is retrospective, and data was collected by chart review where complete information about variables of interest was not available for all of the study population. Due to this, we were unable to distinguish the origin of stroke or TIA as atherosclerotic or cardioembolic.

Thank you again for the time and thoughtful input you have given us on our manuscript,

Michael P Dunleavy, M.D.
